# Global biogeographic regions for ants have complex relationships with those for plants and tetrapods

Runxi Wang [1] ✉, Jamie M. Kass [2,3], Chhaya Chaudhary [1,4], Evan P. Economo [2] & Benoit Guénard [1]

On a global scale, biodiversity is geographically structured into regions of biotic similarity. Delineating these regions has been mostly targeted for tetrapods and plants, but those for hyperdiverse groups such as insects are relatively unknown. Insects may have higher biogeographic congruence with plants than tetrapods due to their tight ecological and evolutionary links with the former, but it remains untested. Here, we develop a global regionalization for a major and widespread insect group, ants, based on the most comprehensive distributional and phylogenetic information to date, and examine its similarity to regionalizations for tetrapods and vascular plants. Our ant regionalization supports the newly proposed Madagascan and Sino-Japanese realms based on tetrapod delineations, and it recovers clusters observed in plants but not in tetrapods, such as the Holarctic and Indo-Pacific realms. Quantitative comparison suggests strong associations among different groups —plants showed a higher congruence with ants than with tetrapods. These results underscore the wide congruence of diverse distribution patterns across the tree of life and the similarities shared by insects and plants that are not captured by tetrapod groups. Our analysis highlights the importance of developing global biogeographic maps for insect groups to obtain a more comprehensive geographic picture of life on Earth.

No two species in the world have the exact same geographic distribution, but many species share similar distributions which can be clustered into discrete biogeographic regions[1–3]. Biogeographic regionalization provides a central framework for understanding these patterns since the 19th century[4–6] and is the basis of modern conservation planning[7–10]. The global biogeographic regions for terrestrial animals and plants have been studied individually[11], i.e., zoogeographic[1,5,12–14] and phytogeographic regions[15–18], because of their distinct ecologies and evolutionary histories. For example, the greater mobility of animals reduces their dependence to local environments compared to plants[19], while plant distributions have been shaped by more ancient historical events[15,17].

However, global regionalizations for terrestrial animals have focused mainly on tetrapods (amphibians, birds, mammals, and reptiles)[1,12–14], neglecting the hyperdiverse insects, which constitute the large majority of animal diversity[20–22]. Due to their much smaller body size, insects are more dependent on microscale environments than tetrapods[23–25] and have a longer shared evolutionary history with plants than tetrapods[19,26]. More importantly, plant-insect interactions, especially mutualisms (e.g., pollination, seed dispersal, defense),

[1]School of Biological Sciences, The University of Hong Kong, Kadoorie Biological Sciences Building, Pok Fu Lam Road, Hong Kong SAR, China. [2]Biodiversity and Biocomplexity Unit, Okinawa Institute of Science and Technology Graduate University, Onna, Okinawa, Japan. [3]Macroecology Laboratory, Graduate School of Life Sciences, Tohoku University, Sendai, Miyagi, Japan. [4]Alfred Wegener Institute, Helmholtz Centre for Polar and Marine Research, Bremerhaven, Germany. ✉e-mail: runxiwg@connect.hku.hk

promote reproduction, dispersal, and colonization[27–30], which have played important roles in diversification for both groups[31–34]. Thus, the spatial congruence of biogeographic regions for plants and insects may be stronger than those with tetrapods, but this hypothesis remains untested due to major knowledge gaps in global regionalizations for insect groups.

Limitations in distributional and phylogenetic information (i.e., Wallacean and Darwinian shortfalls)[2,35,36] hamper the development of a robust global regionalization for most insect taxa. Ants (Hymenoptera, Formicidae) are progressively becoming an exception, being one of the few insect groups with relatively sufficient distributional data[37]. Recent progress in distribution modeling[38] and phylogenetic estimates[39–41] has helped to tackle knowledge shortfalls for ants and has led to improved regionalizations for broad spatial scales[42]. Although most insect species are not yet explored or described[20,22], genus-level taxonomies are often more well-documented, as is the case for ants (Supplementary Fig. 1). Moreover, genus-level datasets allow the inclusion of morphospecies records for genera whose taxonomic knowledge may be particularly deficient, making them attractive candidates for use in regionalizations.

Here, we estimate the global biogeographic structure of ants based on their geographic distributions and evolutionary relationships. We consider both the genus and species levels to provide a complementary, robust, and comprehensive delineation, because distributions at the genus-level may fail to capture more recent ecological and evolutionary processes that shape species' regionalization[43]. Then, we used this regionalization in a first test of the biogeographic congruence among insects and plants and compared it with the congruence between tetrapods and plants. Ants are comparable to or even exceed most tetrapod groups in distribution range (near-globally spread)[37], known species richness (~15,900 described species and subspecies)[44] and ecological dominance in the many ecosystems they inhabit[45–47]. They have diverse interactions with plants, both directly through seed dispersal (with >11,000 angiosperms species) to facultative and obligate protection mutualisms[48,49] to indirect interactions through sap-sucking insects[49], which have resulted in associations regarding patterns of species distribution[50,51], diversification[33,52–55] and richness[56]. These interactions may also lead to a stronger biogeographic congruence between ants and plants on a global scale compared to that between tetrapods and plants. Our results show that the spatial association between ants and plants is stronger than between plants and any tetrapod group. Furthermore, the correlations between ants and various tetrapod groups are also strong, suggesting complex biogeographic relationships among different taxa and the value of including insect groups in global bioregionalizations.

## Results

### Global biogeographic structure of ant biodiversity
The hierarchical clustering analysis showed a strong regionalization of global ant genera based on phylogenetic dissimilarity (Pβsim) and delineated 21 distinct biogeographic regions belonging to 7 larger realms (Fig.1a, b). This biogeographic classification explained most of the Pβsim variance (80% and 90% at the levels of realm and region, respectively, Supplementary Fig. 2). Our analyses recovered two main groups: (1) the combination of the Neotropical and Southern North American realms, and (2) the combination of the Holarctic and Paleotropical realms, the latter consisting of the Indo-Pacific and Australian realms in the east and the Afrotropical and Madagascan realms in the west. In contrast to the discrete groups identified by clustering analysis, the ordination analysis identified affinities between regions on continuous axes and thus showed detailed biogeographic transitions across large realms (Fig.1c, d). For example, affinities between regions were detected around the Gulf of Mexico (SNA1 and

NT1), in central Asia (HA3 and IP3), and in northern Australia (IP1 and AU1) (Fig.1d).

At the species-level, we observed a similar spatial pattern of phylogenetic turnover (Supplementary Fig. 3), but the same major divisions as observed at the genus-level were not recovered: the Southern North American realm was grouped with the Holarctic realm instead of the Neotropical realm, and two more biogeographic realms (Tethyan and Sino-Japanese) were identified between the Holarctic and Paleotropical realms (Fig. 2). The species-level delineation also refined biogeographic regions in the Neotropical and Afrotropical realms. However, for the Southern North America and Indo-Pacific (eastern part) realms, we observed a weaker regionalization (i.e., with fewer biogeographic regions) at the species-level than at the genus-level. Some biogeographic boundaries also shifted; for example, the dispersal barrier separating the ant species in New Guinea from the rest of the Indo-Pacific regions (Lydekker Line) was shifted to the west of Seram Island when using the genus-level delineation (Supplementary Fig. 12). The species-level delineation explained a lower proportion of variance (68% and 81% for realms and regions, respectively, Supplementary Fig. 4) than the genus-level results.

We tested the robustness of the ant biogeographic structure using alternative turnover metrics based on taxonomic dissimilarity, species occurrence data, and alternative clustering algorithms. Taxonomic and phylogenetic turnover were strongly correlated at the genus-level but not at the species-level (Supplementary Fig. 3). However, the species-level taxonomic delineation still recovered a similar structure to the results from the phylogenetic approach and at the genus-level (Supplementary Fig. 5-7). Sensitivity tests based on species occurrence records and alternative clustering algorithms supported the robustness of the results based on range estimates (Supplementary Fig. 8-11), and biogeographic classifications based on range estimates also showed better performance (i.e., the highest variance explained) than delineations based on occurrence data.

### Pairwise comparisons across taxa
The ant biogeographic structure we delineated based on phylogenetic dissimilarity and equal-area geographic units was comparable to previously reported quantitative schema of tetrapods[1,12,13] and vascular plants[15] (see Methods for more details). We quantified the total spatial association (i.e., V measure, $V\beta$) of the phylogenetic regionalizations. Pairwise comparisons showed the highest spatial association between ants and reptiles ($V\beta = 0.80$) and the lowest between vascular plants and amphibians ($V\beta = 0.65$) (Fig. 3 and Supplementary Fig. 13). Vascular plants show higher biogeographic congruence with ants ($V\beta = 0.75$) than with tetrapod groups ($V\beta = 0.65–0.70$). The statistical tests and standardized effect size (SES) based on randomization analysis suggested that only ants and birds showed significant congruence ($p < 0.05$) to vascular plants in biogeographic structure than expected by chance, and vascular plants had the highest SES in spatial association with ants (Fig. 3).

We also quantified spatial patterns of homogeneity between ants and other taxa, which indicate whether a biogeographic region of the focal taxon tends to be nested within a region of the compared taxon (i.e., high homogeneity) or divided by multiple regions of the compared taxon (i.e., low homogeneity). The Northern Hemisphere showed lower regionalization homogeneity than the Southern Hemisphere, while the most congruent biogeographic regions between ants and non-ant taxa were the Madagascan and Oceanian regions (Fig. 4 and Supplementary Fig. 13). Ant regionalizations also supported the uniqueness of the southeastern North American, southern Afrotropical, and southeastern Australian regions at the species-level, which were present for vascular plants but absent for tetrapods (Fig. 4a, d). However, the ant biogeographic structure did not support the delineation of the Amazonian and Guineo-Congolian regions that regionalizations of other taxa suggested (Fig. 4f, j).

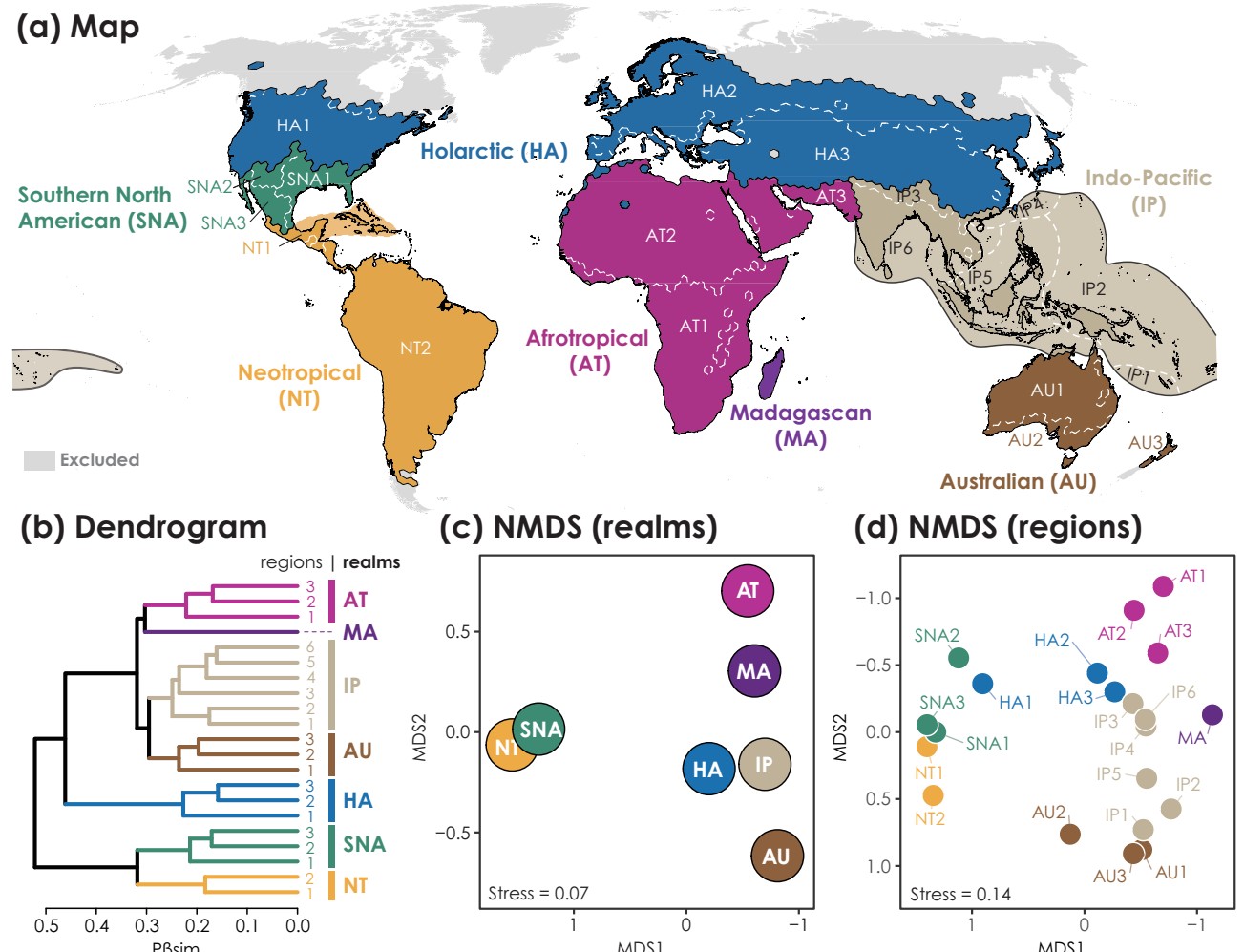

**Fig. 1 | The global phylogenetic biogeographic classification for 267 ant genera shows 7 realms and 21 regions.** Ant biogeographic realms and regions of the world are shown in **a**, and the associated dendrogram resulting from hierarchical clustering based on phylogenetic turnover (Pβsim) across $5 \times 10^4\,km^2$ hexagons is shown in **b**. Ordination results using nonmetric multidimensional scaling (NMDS) show relationships among realms in **c** and regions in **d**. Black solid lines and white lines (dashed on the ocean) delineate the borders of realms and regions, respectively. Colors used to characterize particular realms in maps correspond to those in dendrogram and ordination plots. Areas without sufficient data or invalid biogeographic units are indicated in gray. The map in **a** is in the Robinson projection system, and the dendrogram in **b** used the unweighted pair-group method using arithmetic average (UPGMA).

## Discussion

Here, we derive a quantitative global classification for ants, a major insect group, consisting of 21 distinct biogeographic regions, presenting a first opportunity to compare global regionalization of insect biodiversity with those for terrestrial vertebrates and plants. As we observed a significant and higher congruence between plants and ants than between plants and tetrapods, we demonstrate that the currently acknowledged similarities between zoogeographic and phytogeographic regions[19] might be underestimated. Our results also suggest varied level of associations between ants and different tetrapod groups. These results highlight the importance of including more diverse taxonomic groups in mapping terrestrial biodiversity patterns. To address knowledge shortfalls for insects, we also used multiple approaches and taxonomic scales to account for both spatial and taxonomic incompleteness, and the genus-level approach provided satisfactory results for large-scale classification. Although the regionalization of this large insect family with much ecological and evolutionary diversity has significantly expanded our knowledge, it nonetheless cannot be fully representative of the diversity of all insects. To expand our understanding of insect regionalizations, the approach we demonstrated for ants can be used for other insect groups to understand global patterns, even with incomplete species knowledge.

The strong similarity between ant and plant regionalizations reveals shared ecological and evolutionary processes, particularly highlighting the potential role of biotic interactions. Both abiotic factors[56,57] and biotic interactions[33,53,55] might play important roles, although their relative importance may vary across different regions and biomes. A remarkable diversity of plants with elaiosomes and extrafloral nectaries, structures particularly important in ant-plant mutualisms, are observed in spatial clusters that are shared by both ants and plants but absent in tetrapods, such as Holarctic and Indo-Pacific realms and the Cape region[49,54,56]. In fact, birds and mammals also play important roles in seed dispersal which can expand plant distribution ranges[30], especially through long-distance dispersal[58], and thereby, promote plant speciation[26,59,60]. One possible reason that plants showed weaker spatial associations with birds than ants is that higher extinction rates for plant lineages are observed in bird- and mammal-dispersed groups compared to ant-dispersed groups[59]−this is thought to be because the movement and storage of seeds by ants may help protect seeds from predators and environmental stress[26,48,61].

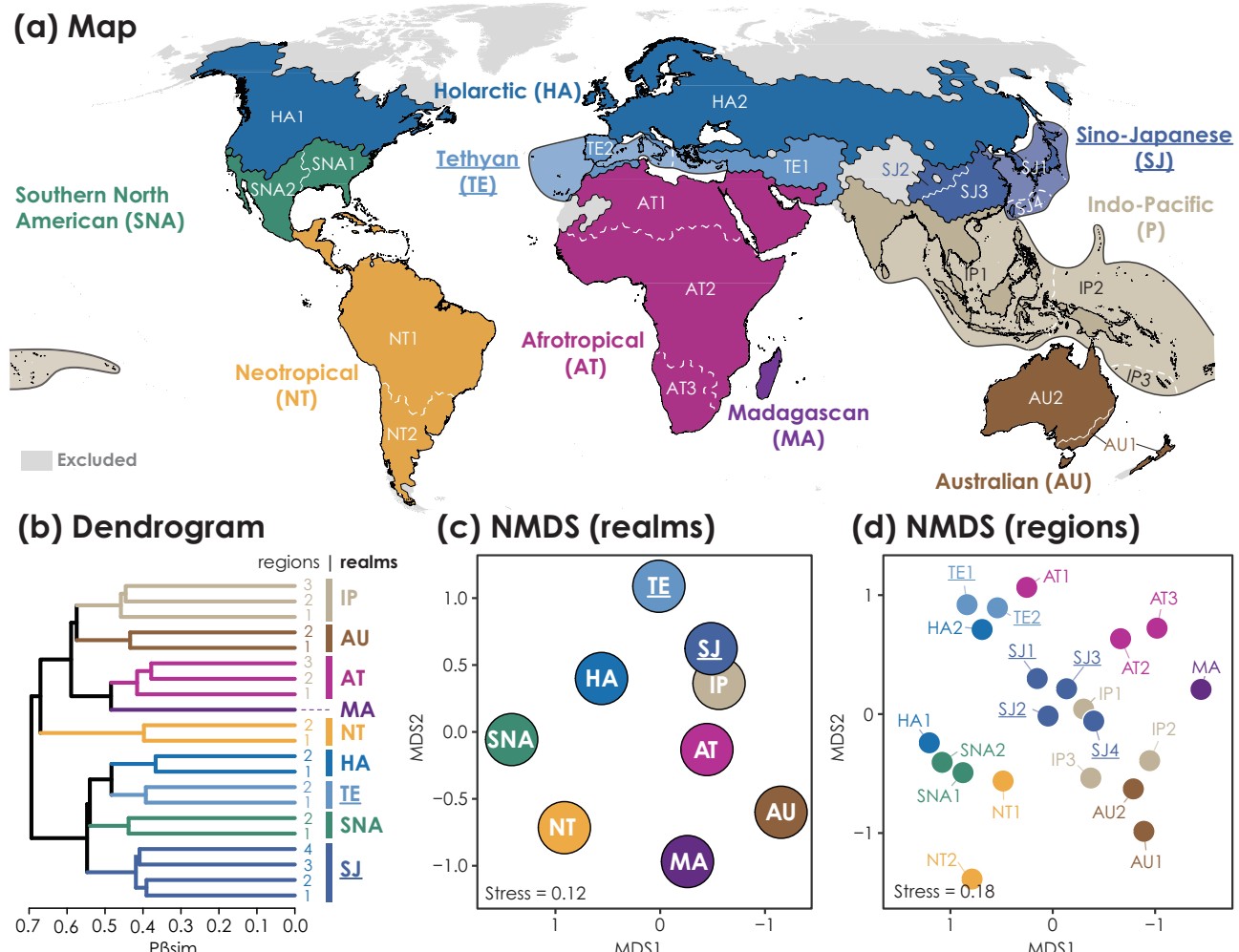

**Fig. 2 | The global phylogenetic biogeographic classification for 13,494 ant species shows 9 realms and 21 regions.** Ant biogeographic realms and regions of the world are shown in **a**, and the associated dendrogram resulting from hierarchical clustering based on phylogenetic turnover (Pβsim) across $5 \times 10^4\,km^2$ hexagons is shown in **b**. Ordination results using nonmetric multidimensional scaling (NMDS) show relationships among realms in **c** and regions in **d**. Two newly defined species-level realms and their affiliated regions are underlined. Black solid lines and white lines (dashed on the ocean) delineate the borders of realms and regions, respectively. Colors used to characterize particular realms in maps correspond to those in dendrogram and ordination plots. Areas without sufficient data or invalid biogeographic units are indicated in gray. The map in **a** is in the Robinson projection system, and the dendrogram in **b** used the unweighted pair-group method using arithmetic average (UPGMA).

The remarkable associations between taxonomic groups highlight that ecological and evolutionary links should help us reconsider the congruence between zoogeographic and phytogeographic regions. While the ant regionalization does not provide an exclusive picture of insect biogeography, its strong congruence with phytogeographic regions is unlikely to be an exception. Diversifications of major insect clades, such as Hemiptera, Hymenoptera, Coleoptera, and Lepidoptera, show close associations with the rise of angiosperms[26,32,62–64]. Besides mutualism, other highly specialized relationships with plants like diet specialization are also widely observed in these key insect clades[24,32]. The distributions and regionalizations of these herbivorous insects are likely influenced by the ranges of their host plants[65], but the degree to which this is so may be unclear for some groups. Moreover, the biogeography of more ancient insect families can also reflect ancient tectonic history such as the breakup of Gondwana[66]. Thus, if we take into account the majority of insects, we should be able to observe more similarities in global regionalization between insects and plants.

The ant regionalization also shows high congruence with many tetrapod groups, which suggests that forces shaping zoogeographic regions are deeply shared across taxa, such as the long-term isolation of islands, contemporary physical barriers, and tectonic activity[42,67,68]. However, varied levels of congruence between ants and tetrapod groups were observed, which can possibly be explained by their distinct ecological characteristics[12,68]. The highest similarity between ants and reptiles may be linked to their shared affinities to arid climates[12,69,70], while amphibians show the lowest similarities with ants and other tetrapod groups, due to their reliance on freshwater habitats[12]. If ecological attributes such as life history and climatic sensitivity play important roles in shaping biogeographic structure[12,68], perhaps biogeographic incongruence can be expected to some extent between insect and tetrapod groups, considering the extremely diverse life histories and high sensitivities to microscale climate that insects have[24].

Our regionalization is in agreement with recent updates for the global biogeographic structures of tetrapods[12,13]. For example, the Madagascan and Sino-Japanese realms are also recognized in our ant regionalization. Additionally, our results also support that the fauna in Melanesia and other Pacific islands (except New Zealand) are more similar to the Oriental than the Australian fauna[71], and they also

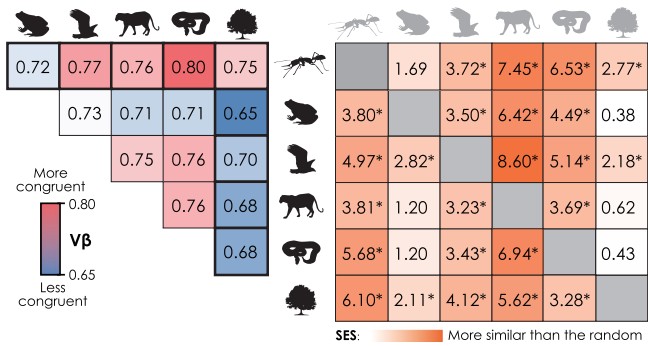

**Fig. 3 | Congruence in spatial association of global regionalizations among taxa at the species-level.** The V-measure ($V\beta$) is an area-weighted harmonic mean of homogeneity between two regionalization schemes, where higher values mean stronger spatial associations. Higher standardized effect size (SES) means that the similarity between the observed biogeographic structure of taxa on the row and column is higher than would be expected by random regionalizations of taxa on the column, with * indicating two-sided p value < 0.05. Comparisons based on the regionalization of ant genera show similar patterns, and details can be found in Supplementary Fig. 13.

support the affinity between the Saharo-Arabian and the rest of the Afrotropical realm[12,13]. Many newly defined regions are considered transition zones that preserve the legacy of biotic exchanges between biogeographic realms throughout geologic history, thereby showing a degree of biogeographic complexity[1,3,72]. Our regionalization provides a powerful framework for understanding the complex biogeographic history of these areas through the lens of ants.

Our analysis is based on the most comprehensive distributional and phylogenetic knowledge databases assembled for ants to date, and we demonstrated that our results are robust to variations in distributional data, dissimilarity matrices, and clustering algorithms. One counterintuitive pattern is the stronger regionalization of genera than species in realms like the Southern North American and Indo-Pacific realms. This may highlight the role of less dispersive and geographically restricted clades, where the species-level regionalization might be dominated by large clades such as *Pheidole* and *Camponotus*, which have the most species. Differences observed between the genus- and species-level regionalizations of ants may also reveal the various historical processes driving biogeographic structure at different taxonomic levels[43]. For example, the dispersal barrier for ant fauna between Sulawesi and New Guinea defined at the genus-level is located in deeper straits compared to the boundary recognized at the species-level (Supplementary Fig. 12), suggesting potential colonization events

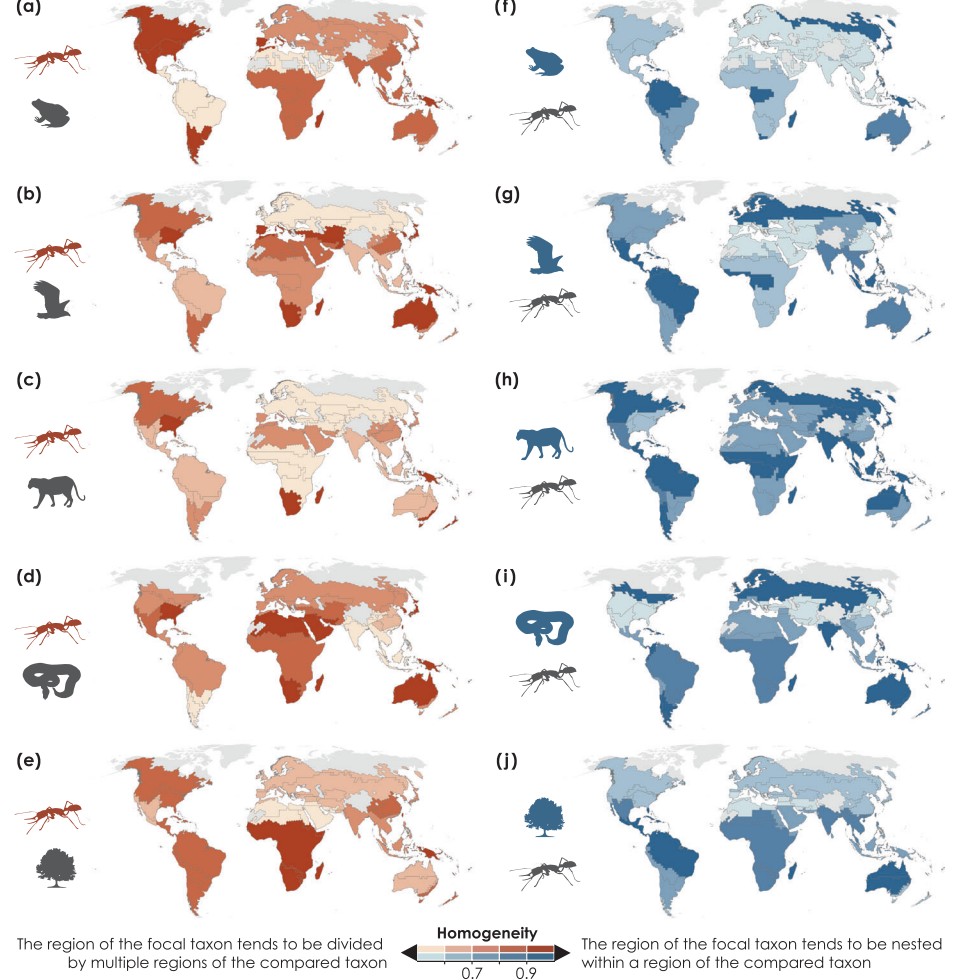

**Fig. 4 | The spatial congruence and divergence of global regionalizations between ant species and other taxa.** Colors indicate the homogeneity of biogeographic regions for ants (light to dark brown) in relation to schema for amphibians (**a**), birds (**b**), mammals (**c**), reptiles (**d**), and vascular plants (**e**) and the schema of these taxa (light to dark blue) in relation to the ant regionalization (**f–j**). Homogeneity is measured by 1 minus the normalized Shannon entropy. Comparisons based on the regionalization of ant genera can be found in Supplementary Fig. 13.

over recent land bridges[71,73,74]. Likely sampling biases may explain the lack of a dispersal barrier between the Andean and Amazonian ant fauna in our regionalization, although high turnover of ant fauna is still observed (Supplementary Fig. 3). These particularly under-sampled regions are predicted to harbor much unknown ant biodiversity[38]. This highlights the pressing need for continuous sampling and taxonomic study efforts in regions with data shortfalls to improve our knowledge of more detailed biogeographic transitions for insects[75,76].

Without the inclusion of insects, current global knowledge on the geographic structure of terrestrial animal diversity remains largely incomplete, and thus cannot inform the conservation of invertebrates that provide key ecosystem functions and services[24]. Certainly, there is still much work to be done describing new ant species, revising taxonomic classifications, and sampling understudied regions, all of which will help refine and improve our biogeographic delineations. Our regionalization provides the best window yet for global maps of insects by employing range estimates for each genus and species, as well as a large-scale phylogeny for global ant fauna. High associations among the biogeographies of different taxa suggest the possibility of global regionalizations integrating both zoogeographic and phytogeographic regions, but notable spatial divergences driven by ecological and evolutionary differences between them also indicate that responses to global change may differ among distinct clades. Therefore, future research should prioritize filling biogeographic knowledge gaps for a variety of insect groups in order to recognize their spatial structure and define conservation priorities.

## Methods
### Data overview
Incomplete knowledge of taxonomy, geographic distributions, and phylogeny represents a major challenge to understanding patterns of insect biodiversity. We addressed these shortfalls for ants by (1) maximizing the availability of distributional data by including morphospecies at the genus-level; (2) using species distribution models (SDMs) to make range estimates; and (3) using a comprehensive phylogeny to obtain evolutionary information and its associated uncertainty.

### Ant distribution data
We compiled both genus- and species-level distributional information for our regionalization. The Linnean Shortfall is generally strong for insects[2], as nearly >80% of insect species are thought to be undescribed[22]. Taxonomic progress made for ants suggests a high potential for new species discoveries (Supplementary Fig. 1) that also highlight the evolving nature of species-level classification. As taxonomic revisions would greatly influence delineation results, this is an important limitation. In contrast, although new genera are occasionally described[77], genus-level classification has been relatively well-resolved[40] (Supplementary Fig. 1), making the description of ant genera more stable. Moreover, as both nominal species and most morphospecies records can be used to determine the distribution of each genus, this allowed the inclusion of an additional ~350,000 distributional records into our database, in particular for genera that are rarely collected, poorly defined at the species-level or for which misidentifications can be common (e.g. *Nylanderia*[78]). Only a subset of genera that did not have major taxonomic changes in past decades (e.g., morphospecies for *Cerapachys, Gnamptogenys, Pachycondyla, Paratrechina, Stigmatomma*, among others, were excluded due to recent taxonomic splits), or for which previous names could be converted directly (e.g., *Pyramica* to *Strumigenys*), was included in our analyses.

We downloaded 2,503,609 distributional records of ants from the Global Ant Biodiversity Informatics (GABI) database[37], which includes data from the literature (both English and non-English languages), specimen information (e.g., museum and personal collections), and unpublished datasets. Although some regions still have data shortfalls

(e.g., Tibet, the Sahara Desert, west of the Andes Mountains) due to limited sampling and taxonomic effort, the dataset we used here represented the most comprehensive distributional knowledge of extant ants in the world. All subspecies were treated as species for analysis purposes, and the nomenclature was updated up to 1st March 2023 based on the Online Catalog of the Ants of the World (AntCat)[44]. Additionally, we removed records for non-native species and those with invalid geographic information based on a comprehensive and evidence-based framework that detects different stages of invasion and levels of invasion capacity[37,79]. The final dataset we used included georeferenced records for 345 genera and 14,324 species and subspecies.

### Range estimations
We used range estimates from a previous study by Kass et al.[38] for global ant species and performed new estimates for the genus level. In this study, polygonal range estimates for all taxa (species and genera) were made by delineating alpha hulls around occurrence points and buffering by 30 km (when data were limited, buffered convex hulls or point buffers were used). Range estimates for low-data species/genera (<5 occurrence records) were restricted to polygonal range estimates. For species/genera with sufficient data (≥5 occurrence records), species distribution models (SDMs) were also used to make modeled range estimates. These were built using occurrence data and bioclimatic predictor variables (10-arcminute resolution, or ~20 km at the equator) from Worldclim 2.0[80]. As absence data was lacking, models were fit with background data sampled from within the species' polygonal range estimates. Models were fit using the machine-learning algorithm Maxent 3.4.3[81] and tuned for optimal complexity with the R package ENMeval 2.0.0[82]. Models with the highest performance for random cross-validation were selected, based on sequential evaluation criteria (lowest 10-percentile omission rate first, and highest validation AUC second; see Kass et al.[38] for details). These tuned models were then used to predict continuous estimates of environmental suitability over the extents of the polygonal range estimates, effectively constraining range estimations to the limits of the occurrence data. For this study, we made binary range maps for species/genera by thresholding the continuous SDM range estimates with the same 10-percentile suitability values used to calculate omission rates. For low-data species/genera, we simply rasterized the polygon range estimates to 10-arcminute grids.

In order to standardize ant ranges at the appropriate geographic scale, we then projected the binary range maps to an equal-area hexagon grid ($5 \times 10^4$ km$^2$, Behrman projection) covering the Earth's surface. Hexagons are less sensitive to distortion on a large scale than squares, as they are less biased by the edge effect and have equidistant centroids with their neighbors. The resolution we used, comparable to grid cells at 2° (~220 km at the equator) resolution, has previously been employed for other broad-scale studies[1,13,15]. We resampled the gridded binary range maps to the hexagon grid using maximum neighborhood values, resulting in ant taxon presence being assigned to a hexagon if any overlapping 10-arcminute grid cells had predictions of presence. Hexagons with fewer than 5 taxa assigned as present were excluded from later analyses to avoid potential distortion due to small sample sizes in dissimilarity analyses[1]. Ultimately, we estimated ant genus presence in 3420 hexagons and species presence in 3095 hexagons based on binary range maps. All analyses were conducted in R version 4.2.1[83], and we used the R packages raster 3.5-11[84] and sf 0.9-6[85] to process rasters and shapefiles, respectively.

### Ant phylogeny
We derived phylogenetic information from a recently reconstructed large-scale phylogeny of ants[40]. The phylogeny was grafted by 100 backbone trees of 262 terminal clades from the posterior and represented the phylogenetic relationships of >14,000 ant taxa with their

uncertainty. The phylogeny was well-resolved at the clade level (mainly representing the genus-level classification) with the topology of taxa within the terminal clades randomly resolved. Although the detailed topology of the phylogeny at the species-level was still limited, the phylogeny we used here was appropriate for species-level dissimilarity analysis[13,15,86–88] as the pairwise dissimilarity (for details, see Turnover section) is not sensitive to the topological uncertainty in phylogeny[89–91]. Moreover, an empirical comparison suggested that the phylogenetic structure of species assemblages, which was also calculated based on branch length, showed a significant correlation with the assemblage structure at the genus-level[92]. We updated the nomenclature of phylogenetic trees and pruned them based on genus/species lists for our study to exclude some invalid/synonymous taxa. Phylogenetic trees were processed using the R packages geiger 2.0.7[93] and picante 1.8.2[94]. Ultimately, there were 267 genera (77% coverage) and 13,494 species (94% coverage) included in our phylogenetic analyses (details can be found in Data availability).

## Existing global biogeographic schema for tetrapods and vascular plants

We compared ant biogeographic structures with recent schema for terrestrial vertebrates[12,13] and vascular plants[15]. The zoogeographic schema included regionalizations for all tetrapods: amphibians (19 regions), birds (19 regions), mammals (34 regions), and reptiles (24 regions). Although the phylogenetic dissimilarity distance used for vertebrates was quantified by phylogenetic branch count instead of branch length, it did not affect the large-scale patterns of results[13]. The regionalization scheme for vascular plants was delineated into 16 regions, belonging to three phytogeographic kingdoms. The schema for tetrapods and plants were all delineated using the same multivariate analysis and dissimilarity distance metric (Simpson index of dissimilarity) based on phylogenetic trees.

## Regionalization framework

The regionalization analysis consisted of three steps: (1) delineating a global biogeographic classification for ants; (2) describing the relationships between biogeographic regions and their transitions; and (3) comparing the biogeographic structure of ants with the regionalizations for vertebrates and plants. To make even comparisons, we used the same quantitative methodological framework as was used for the global schema for tetrapods and plants. Specifically, we delineated the biogeographic classification using a hierarchical clustering analysis based on dissimilarity distance metrics, then used an ordination analysis to illustrate relationships between different regions[1,13,15].

We quantified dissimilarity as the turnover component of assemblages, or the replacement of species or evolutionary histories over space[95]. The use of hierarchical clustering maximizes the similarity of assemblages within the targeted biogeographic region and the heterogeneity among different regions in a hierarchy. Alternatively, ordination approaches can map assemblages into a low-dimensional space to aid visualization of their relative positions based on their dissimilarity distances. This approach better illustrates the transition between biogeographic regions, which is difficult to detect with clustering analyses[96]. The recent emergence of new techniques (e.g., network analysis[97]) has led to exciting insights into biogeographic regionalization. However, our focus is not just to delineate the biogeographic regions of ants but also to produce a comparable scheme to existing ones. Also, as these new methods have difficulty integrating phylogenetic information[98], they were not considered in this analysis. Nomenclatures of biogeographic realms followed the previous conventions, i.e., using existing names to define similar areas instead of creating new ones[3,6]. Thus, names of biogeographic realms mainly referred to the latest zoogeographic regionalization[13] and other references of ant biogeography[99].

## Turnover

We used Simpson's index of dissimilarity (βsim) to measure the turnover component[95]. The βsim metric is not sensitive to species richness and can be calculated based on both taxonomic composition (i.e., Tβsim) and phylogenetic branches (i.e., Pβsim). Taking the Pβsim as an example, the pairwise value between two assemblages was calculated by:

$$1 - (a/(\min(b, c) + a))$$

where $a$ is the total length of phylogenetic branches shared by two assemblages, and $b$ and $c$ are the length of unique phylogenetic branches of each assemblage[13,100].

After calculating the pairwise Pβsim across 100 posterior trees, we used the median value of Pβsim for each assemblage pair. The Tβsim was calculated as the total number of taxa (genera/species) that two assemblages shared (i.e., $a$ in the above equation) divided by the number of unique taxa in each of the assemblages (i.e., $b$ and $c$ in the above equation). Higher or lower values of βsim indicated more or less dissimilarity between the two assemblages. To map the spatial patterns of distinctiveness of each assemblage, we also calculated the mean value of βsim between the focal assemblage and the rest of the assemblages (at a global scale). The dissimilarity analysis was conducted using the R package betapart 1.5.2[101].

## Clustering and ordination analyses

We tested seven clustering algorithms to classify the biogeographic regions using cophenetic Pearson correlation and the Gower distance to measure the degree of data distortion in models[1]. The algorithms tested were unweighted pair-group method using centroids (UPGMC); Ward's method (WARD); single lineage (SL); complete lineage (CL); weighted pair-group method using arithmetic averages (WPGMA); weighted pair-group method using centroids (WPGMC); and unweighted pair-group method using arithmetic averages (UPGMA). The UPGMA method consistently had the best performance (i.e., the highest cophenetic correlation coefficients with the lowest Gower distance among them; Supplementary Table. 1) and thus was used for further analysis.

We then used the 'elbow' (or 'knee') method to evaluate the optimality of cluster numbers that referred to the biogeographic region-level classification[100]. The 'elbow' method detects the optimal number of clusters by identifying the point of maximum curvature based on the percentage of variance explained by the given number of clusters (up to 30). Only clusters including more than 10 hexagons and clearly aggregated in the space were identified as valid biogeographic regions[13,15]. Biogeographic regions were further grouped into realms to visualize the larger-scale spatial organization of distributions. To be comparable with the realms of vertebrates, the number of biogeographic realms of ants was determined by the 'elbow' point that was smaller than 15 clusters[6,11].

Lastly, we used nonmetric multidimensional scaling (NMDS) to visualize the relationships between biogeographic regions/realms in two dimensions[96,102]. The ant assemblages were ranked based on their pairwise distance (i.e., βsim) by 200 random starts to find the stable solution and avoid local minima. The stress value was used to measure the goodness of ordination fit and calculated by the sum of the squared differences between the fitted and original distances, where a lower stress value indicated better fit for the ordination. Ordination results were rotated to maximize the congruence of their relative positions between ordination space and geographical space. We used the R package vegan 2.5.6[103] to perform the clustering and ordination analyses and visualized clustering dendrograms and biogeographic maps using the R package phyloregion 1.0.8[86].

## Sensitivity analyses

We also tested how sensitive our delineations were to different alternative dissimilarity metrics, distributional data, and clustering algorithms. We calculated turnover and performed clustering analyses based on alternative metrics: dissimilarity distance based on species compositions (i.e., Tβsim) and the occurrence records of ant genera in hexagons (2774 hexagons included). We compared the pairwise distance matrices of Pβsim and Tβsim and pairwise matrices between the estimated and raw data. We assessed the strengths of these relationships using the Mantel correlation test in the R package vegan[103], and tested spatial congruence with a modified $t$ test that corrects the degrees of freedom to control for spatial autocorrelation[104] using the R package SpatialPack[105]. Additionally, we used an unbiased clustering algorithm to test the sensitivity of our βsim matrices to zero values and variation in the input order of data[106] using the R package recluster 2.8[106,107].

## Congruence of different biogeographic schemes

We performed pairwise comparisons between the regionalization of ants and schema of other taxa. We quantified the degree of spatial association with the V-measure ($V\beta$) using the R package saber 0.4.3[108]. $V\beta$ is an area-weighted harmonic mean of homogeneity of two regionalization schemes, where a higher $V\beta$ value indicates a stronger net spatial association. Then we used a randomization approach to evaluate whether the $V\beta$ value was significantly higher than expected by chance following Falaschi et al.[12]. This involved comparing observed $V\beta$ between two regionalizations to null models of $V\beta$, which were calculated based on 999 random regionalizations generated for each taxon in each pairwise comparison.

Each pairwise comparison thus yields two $p$-values: one represents the proportion of comparisons between taxon A and null models of taxon B where $V\beta$ was higher than their true regionalization, while the other represents the proportion of comparisons between taxon B and null models of taxon A with higher $V\beta$. To quantify and compare the degree of biogeographic similarities between different pairs of regionalizations, we calculated the SES, which divided the differences between the observed $V\beta$ and mean $V\beta$ obtained from null models by the standard deviation of null distribution of $V\beta$ within each pair. The 'voronoi' function of the R package dismo 1.3-3[109] was used to generated randomized regionalizations.

We also quantified the homogeneity of each biogeographic region per taxon with respect to the region of the compared taxon in each regionalization pair. The homogeneity value indicates whether a biogeographic region of the focal taxon tends to be nested within a region of a compared taxon (i.e., high homogeneity), or alternatively divided by multiple regions of the compared taxon (i.e., low homogeneity).

## Reporting summary

Further information on research design is available in the Nature Portfolio Reporting Summary linked to this article.

## Data availability

The ant distribution data are available from the GABI database[37], and the reconstructed phylogenies are from Economo et al.[40]. Continuous range estimates for ant genera and species are available from the supplemental data section of Kass et al.[38] at https://datadryad.org/stash/dataset/doi:10.5061/dryad.wstqjq2pp. The final dataset used for regionalization analysis has been deposited in Figshare [https://doi.org/10.6084/m9.figshare.25011866]. The zoogeographic regions of amphibians, birds and mammals[13] were accessed from https://macroecology.ku.dk/resources/wallace/, reptiles[12] were accessed from https://doi.org/10.6084/m9.figshare.19844755 and the phylogeographic regions for vascular plants were obtained from Carta et al.[15], available at https://github.com/spiritu-santi/Floristic-Kingdoms/tree/main/shapefiles.

## Code availability

The R scripts used for regionalization analyses have been deposited in Figshare [https://doi.org/10.6084/m9.figshare.25011866]. The script used for V-measure randomization analysis was modified from Falaschi et al.[12], available at https://doi.org/10.6084/m9.figshare.19844755.

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

## Acknowledgements

B.G. acknowledges funding support from an Early Career Scheme Grant from the Research Grants Council (ECS-27106417) of the Hong Kong Government. The authors would like to thank Michael D. Weiser, Clinton N. Jenkins, Nathan J. Sanders, and Robert R. Dunn for early discussions and work on global ant biogeography. This project was funded by the Faculty of Science RAE Improvement Fund (2019) and from the 39th Post-Doctoral Fellowship/RAP Scheme from The University of Hong Kong to B.G.

## Author contributions

R.W., C.C., and B.G. conceived and designed the project and compiled the datasets. R.W., J.M.K., and C.C. performed the analyses under the guidance of B.G. and E.P.E. R.W. wrote the manuscript, and J.M.K, E.P.E., and B.G. edited the manuscript. All authors reviewed and approved the manuscript.

## Competing interests

The authors declare no competing interests.
