## [Peer Review File · Nature Communications]

Global biogeographic regions for ants have complex relationships with those for plants and tetrapodsREVIEWER COMMENTS

Reviewer #1 (Remarks to the Author):

I found this manuscript to be a very useful contribution, an important development in the field of bioregionalization. As the authors point out, there are relatively few invertebrate-based exercises due to data incompleteness, a problem the authors get around fairly elegantly – and they discuss how this may yet affect their results. They also explore a broad range of methods, both in the actual regionalization analyses, and in assessing the levels of agreement between their ant-based scheme and vertebrate as well as plant ones. The writing is good, and so are the displays.

The main improvements I suggest emanate largely from my own work. Asking the authors to cite these would of course be conflicted. I am simply mentioning these because I am familiar with the work, and the authors should only cite these where they believe this is genuinely deserving.

1) Another example of an insect-based global regionalization exercise.

https://www.researchgate.net/publication/360725449_Analysis_of_the_diversity_and_distributional_patterns_of_coleopteran_families_on_a_global_scale

2) Lines 201-202: I agree that microclimatic pockets make a difference here; in fact, I think these are key to defining some regional invertebrate faunas. To a lesser extent this is also the case with plants, but very seldom with vertebrates. I tend to think this is the primary reason why ants and plants are more similar. Direct links between ants and plants and even between plants and other insects with overall greater level of host specialization (lines 164-189) may be overstated here and in the literature in general, despite the long co-evolution over geological time. It is true that in tropical rainforest plant diversity directly drives insect diversity, as illustrated in the work of Novotny et al.

<https://www.science.org/doi/abs/10.1126/science.1129237?versioned=true>

, but in other vegetation types we have shown that the plant-insect diversity relationship is overwhelmingly driven by similar responses to environmental factors (in fact, the only biome where we picked up any effect of a direct relationship was a thicket type closest to rainforest among our studied ecosystems).

<https://www.sciencedirect.com/science/article/abs/pii/S1055790308003102>

These observations on diversity relationships would likely hold for regionalization on a global scale, although finer scale implications would differ. In a global regionalization perspective, tropical rainforest is important, but patterns observed there should not dictate general conclusions.

3) The figure with levels of agreement (fig 3) is possibly too complicated, including different measures. I would just present one measure, and maybe relegate the other pane to Supplementary. Up to the editor though.

Serban Proches

Reviewer #2 (Remarks to the Author):

Ants have garnered substantial attention (relative to most other groups of insects) among biologists and the general public alike due to their ubiquity, ecological roles, social behaviors, and other traits. Over the past few decades, researchers have made considerable progress in increasing our understanding of ant taxonomy, phylogenetic history, and distributions. The authors of this study leverage these recent advances to analyze the global regionalization for ants and compare their results to more well-studied tetrapod and plant groups.

It is quite the achievement to conduct these types of sophisticated biogeographical analyses on ants because they require very detailed and “comprehensive” data (in quotes because our knowledge of any insect group, even the relatively well-known ants, is undoubtedly woefully incomplete). The methodology is rigorous and presented in sufficient detail. They found higher biogeographic congruence of plants with ants compared to tetrapods, which makes sense given the ecological interactions plants can have with ants. This conclusion is appropriately supported by the presented results.

This study is also important because, I believe for the first time, it includes a major insect group in these types of modern biogeographical analyses. The authors rightfully point out that similar trends may be identified in the future for other insect groups with close associations with plants.

Detailed comments:

Line 29: Sentence ends in an incomplete phrase. Suggest replacing “, but” with “and also” so that it reads “... of life and also the similarities...”

Line 31: Even if all insect groups are somehow included, our picture would still be very “biased” because many other organismal groups are still missing. Suggest replacing “unbiased” with “more complete”.

Lines 60-64: I think this provides a good justification for including genus-level data, but it begs the question of how valid “genera” are as your unit of analysis. Particularly, since evolutionary history is a fundamental component of the analyses, doesn't this assume that genera are valid evolutionary units (e.g, monophyletic)? What if members of a genus are, in reality, all not closely related to each other? I think brief recognition of this issue (or rebuttal if I am wrong) is warranted.

Line 74: the word “ecologically” should read “ecological”.

Lines 88-91. These lines report the two main groups recovered in the genus-level analysis. However, these two main groups are NOT recovered in the species-level analysis. This is hinted at below (lines 98-104) by referring to “weaker regionalization...at the species-level than at the genus level”. But I think it should be clearly stated that the species-level analysis did not recover the same main groups as the genus-level one.

Lines 127-8: I think the reference to Supplementary Fig. 12 is wrong? My version of Supp Fig.12 is the one focused only on ants (and not non-ants) only in the Wallacean region.

Line 132: I would reference Fig. 3 at the end of this sentence, since this is where these important results are provided.

Line 140: Same problem with Supp Fig. 12.

Line 149: Slight grammatical issue – “those” (plural) refers to “regionalization (singular).

Line 151: The authors refer to “currently reported values of similarity between zoogeographic and phylogeographic regions”. Please provide citations for these reports.

Line 164: Delete “the” to now read “...reveals shared ecological...”

Line 216: Here you claim to have “demonstrated that our results are robust to variations in taxonomic levels”. Yet you then immediately discuss “one counterintuitive pattern is the stronger regionalization of genera than species in realms like...” and “differences observed between the genus- and species-level regionalizations of ants...”. Isn’t this because you got different results at different taxonomic levels? How do you know these differences are “robust”, instead of one or the other not being simply wrong or misleading?

Line 226: Supp Fig. 13 is cited, but I think this one is actually supposed to be Supp Fig. 12?

Line 282: How was it determined which species were “non-native” and from exactly which localities they were non-native?

Lines 331-2: Incomplete sentence.

Fig. 4 Legend, last sentence: Again, I think Supp Fig. 12 is not the correct figure to reference here.

REPLIES TO REVIEWERS' COMMENTS

Reviewer #1 (Remarks to the Author):

I found this manuscript to be a very useful contribution, an important development in the field of bioregionalization. As the authors point out, there are relatively few invertebrate-based exercises due to data incompleteness, a problem the authors get around fairly elegantly – and they discuss how this may yet affect their results. They also explore a broad range of methods, both in the actual regionalization analyses, and in assessing the levels of agreement between their ant-based scheme and vertebrate as well as plant ones. The writing is good, and so are the displays.

Response: Thank you very much for your time and effort reviewing our manuscript, and also for your positive comments. We have revised the manuscript and cited the important references you suggested. We provide point-by-point responses to your comments below.

The main improvements I suggest emanate largely from my own work. Asking the authors to cite these would of course be conflicted. I am simply mentioning these because I am familiar with the work, and the authors should only cite these where they believe this is genuinely deserving.

1) Another example of an insect-based global regionalization exercise.

https://www.researchgate.net/publication/360725449_Analysis_of_the_diversity_and_distributional_patterns_of_coleopteran_families_on_a_global_scale

Response: Thank you for the suggestion. Indeed, this paper is an important but missing reference in our manuscript. We have now added it in the main text (line 47) and changed the relevant statement as well (line 22):

Here, we developed ~~the~~ a first global regionalization for a major and widespread insect group, ants, based on the most comprehensive distributional and phylogenetic information to date...

Also (lines 151-153):

Here, we derive the ~~first~~ first quantitative global classification **for ants, a major insect group,** consisting of 21 distinct biogeographic regions ~~for a major insect group, the ants, and this classification~~ **presenting** a first opportunity to compare global regionalization of insect biodiversity with **those** ~~those that of~~ **for** terrestrial vertebrates and plants.

2) Lines 201-202: I agree that microclimatic pockets make a difference here; in fact, I think these are key to defining some regional invertebrate faunas. To a lesser extent this is also the case with plants, but very seldom with vertebrates. I tend to think this is the primary reason why ants and plants are more similar. Direct links between ants and plants and even between plants and other insects with overall greater level of host specialization (lines 164-189) may be overstated here and in the literature in general, despite the long co-evolution over geological time. It is true that in tropical rainforest plant diversity directly drives insect diversity, as illustrated in the work of Novotny et al. <https://www.science.org/doi/abs/10.1126/science.1129237?versioned=true>

, but in other vegetation types we have shown that the plant-insect diversity relationship is overwhelmingly driven by similar responses to environmental factors (in fact, the only biome where we picked up any effect of a direct relationship was a thicket type closest to rainforest among our studied ecosystems).

<https://www.sciencedirect.com/science/article/abs/pii/S1055790308003102>

These observations on diversity relationships would likely hold for regionalization on a global scale, although finer scale implications would differ. In a global regionalization perspective, tropical

rainforest is important, but patterns observed there should not dictate general conclusions.

Response: Thanks for the comment. We agree that the abiotic factors shared by insect and plant distributions are important and the importance of biotic interactions may vary across different regions. Furthermore, a recent study about co-diversification between ants and plants also highlight the role of plants that provides diverse niches for ants. Therefore, we have cited the reference and made the following revision (lines 168-171):

The strong similarity between ant and plant regionalizations reveals ~~the~~ shared ecological and evolutionary processes, particularly highlighting the potential role of biotic interactions. Both abiotic factors^{56,57} and biotic interactions^{33,53,55} might play important roles, although their relative importance may vary across different regions and biomes.

3) The figure with levels of agreement (fig 3) is possibly too complicated, including different measures. I would just present one measure, and maybe relegate the other pane to Supplementary. Up to the editor though.

Serban Proches

Response: Thank you for your suggestion. Fig.3 presented the measurement of pairwise spatial congruence among different regionalizations using the V-measure and its standardized effect size. We acknowledge that the information in Fig.3 is perhaps complicated, but it provides necessary and complementary information for the same measurement, i.e., the value of V-measure and its statistical significance. Therefore, we believe retaining both sets of information are important here.

Reviewer #2 (Remarks to the Author):

Ants have garnered substantial attention (relative to most other groups of insects) among biologists and the general public alike due to their ubiquity, ecological roles, social behaviors, and other traits. Over the past few decades, researchers have made considerable progress in increasing our understanding of ant taxonomy, phylogenetic history, and distributions. The authors of this study leverage these recent advances to analyze the global regionalization for ants and compare their results to more well-studied tetrapod and plant groups.

It is quite the achievement to conduct these types of sophisticated biogeographical analyses on ants because they require very detailed and “comprehensive” data (in quotes because our knowledge of any insect group, even the relatively well-known ants, is undoubtedly woefully incomplete). The methodology is rigorous and presented in sufficient detail. They found higher biogeographic congruence of plants with ants compared to tetrapods, which makes sense given the ecological interactions plants can have with ants. This conclusion is appropriately supported by the presented results.

This study is also important because, I believe for the first time, it includes a major insect group in these types of modern biogeographical analyses. The authors rightfully point out that similar trends may be identified in the future for other insect groups with close associations with plants.

Response: Thank you very much for your time and effort reviewing our manuscript and for your positive and detailed comments. We have revised the manuscript following your suggestions, modified some statements and corrected any mistakes. We provided point-by-point responses to your comments below.

Detailed comments:

Line 29: Sentence ends in an incomplete phrase. Suggest replacing “, but” with “and also” so that it reads “... of life and also the similarities...”

Response: Thanks for the suggestion, we’ve revised it accordingly.

Line 31: Even if all insect groups are somehow included, our picture would still be very “biased” because many other organismal groups are still missing. Suggest replacing “unbiased” with “more complete”.

Response: Thank you for the suggestion, we agree and have modified the sentence accordingly.

Lines 60-64: I think this provides a good justification for including genus-level data, but it begs the question of how valid “genera” are as your unit of analysis. Particularly, since evolutionary history is a fundamental component of the analyses, doesn’t this assume that genera are valid evolutionary units (e.g. monophyletic)? What if members of a genus are, in reality, all not closely related to each other? I think brief recognition of this issue (or rebuttal if I am wrong) is warranted.

Response: Thanks for your comments. It’s true that the validity of taxonomic units in analysis is very important, and this issue may appear at both the genus- and species-levels. Further, the taxonomic revision at species-level may also lead to taxonomic changes at the genus-level. In fact, we have highlighted this issue in the methods (lines 265-270):

Taxonomic progress made for ants suggests a high potential for new species discoveries (Supplementary Fig.1) that also highlight the evolving nature of species-level classification. As taxonomic revisions would greatly influence delineation results, this is an important limitation. In contrast, although new genera are occasionally described^{78,75}, genus-level classification has been relatively well-resolved⁴⁰ (Supplementary Fig.1), making the description of ant genera more stable.

Both the backbone phylogeny of ants that we used and resolution of a majority of ant genera appears now well established. But we’ve also highlighted this issue in the discussion as below now (line 246):

Certainly, there is still much work to be done describing new ant species, **revising taxonomic classifications**, and sampling understudied regions, **both all** of which will help refine and improve our biogeographic delineations.

Line 74: the word “ecologically” should read “ecological”.

Response: Thanks, we’ve revised this.

Lines 88-91. These lines report the two main groups recovered in the genus-level analysis. However, these two main groups are NOT recovered in the species-level analysis. This is hinted at below (lines 98-104) by referring to “weaker regionalization...at the species-level than at the genus level”. But I think it should be clearly stated that the species-level analysis did not recover the same main groups as the genus-level one.

Response: Thanks for the comment. We agree, and have revised the text (lines 100-103):

At the species-level, we observed a similar spatial pattern of phylogenetic turnover (Supplementary Fig.3), but **the same major divisions as observed at the genus-level were not recovered: the Southern North American realm was grouped with the Holarctic realm instead of the Neotropical realm** and ~~we identified~~ two more biogeographic realms (Tethyan and Sino-Japanese) were identified between the Holarctic and Palearctic realms (Fig.2).

Lines 127-8: I think the reference to Supplementary Fig. 12 is wrong? My version of Supp Fig.12 is the one focused only on ants (and not non-ants) only in the Wallacean region.

Response: Thank you for pointing out this mistake, we've revised.

Line 132: I would reference Fig. 3 at the end of this sentence, since this is where these important results are provided.

Response: Thank you, we've added it.

Line 140: Same problem with Supp Fig. 12.

Response: Thank you for pointing out this mistake, we've revised.

Line 149: Slight grammatical issue – “those” (plural) refers to “regionalization (singular).

Response: Thank you, we've revised.

Line 151: The authors refer to “currently reported values of similarity between zoogeographic and phytogeographic regions”. Please provide citations for these reports.

Response: Thanks for the comment, we're sorry that the statement was inaccurate, and we've revised and added citation as below (lines 153-156):

As we observed a significant and higher congruence between plants and ants than between plants and tetrapods, we demonstrate that the currently ~~reported values of~~ **acknowledged similarity similarities** between zoogeographic and phytogeographic regions¹⁹ might be underestimated.

Line 164: Delete “the” to now read “...reveals shared ecological...”

Response: Thank you, we've revised.

Line 216: Here you claim to have “demonstrated that our results are robust to variations in taxonomic levels”. Yet you then immediately discuss “one counterintuitive pattern is the stronger regionalization of genera than species in realms like...” and “differences observed between the genus- and species-level regionalizations of ants...”. Isn't this because you got different results at different taxonomic levels? How do you know these differences are “robust”, instead of one or the other not being simply wrong or misleading?

Response: Thank you for the comment. Yes, the variations in taxonomic levels do lead to different results, so we've removed this misleading text. In this revised version, we now highlight the differences in regionalizations between the genus- and species-levels, as well as the potential reasons behind them, as suggested in your comments above:

(lines 100-103)

At the species-level, we observed a similar spatial pattern of phylogenetic turnover (Supplementary Fig.3), but **the same major divisions as observed at the genus-level were not recovered: the Southern North American realm was grouped with the Holarctic realm instead of the Neotropical realm** and ~~we identified~~ two more biogeographic realms (Tethyan and Sino-Japanese) were identified between the Holarctic and Palearctic realms (Fig.2).

(line 246)

Certainly, there is still much work to be done describing new ant species, **revising taxonomic classifications**, and sampling understudied regions, ~~both~~ **all** of which will help refine and improve our biogeographic delineations.

Line 226: Supp Fig. 13 is cited, but I think this one is actually supposed to be Supp Fig. 12?

Response: Thank you for pointing out this mistake, we've revised.

Line 282: How was it determined which species were “non-native” and from exactly which localities they were non-native?

Response: Thanks for the comment. Recognizing the extent of native versus non-native ranges in ants is indeed a challenging task and our current knowledge may have uncertainties. To address this, the Global Ant Biodiversity Informatics Project (Guénard et al. 2017), which we used as the source of ant data for this analysis, integrates a vast amount of literature and expert opinions to determine native and non-native records within known distributions, and it is updated regularly based on the latest studies. Non-native records were identified using a comprehensive and evidence-based framework (Wong et al., 2023) that detects different stages of invasion and levels of invasion capacity, including “transported”, “established indoors” and “naturalized” categories. The incorporation of detailed information and evidence-based assessment ensures the reliability in our delimitation of ants' native ranges to the best of our knowledge. We have revised and added citations to improve the transparency of this step as below (lines 288-290):

~~After~~ Additionally, we removed records for non-native species and those with invalid geographic information based on a comprehensive and evidence-based framework that detects different stages of invasion and levels of invasion capacity^{37,80}.

References:

- Guénard, B., Weiser, M. D., Gomez, K., Narula, N., & Economo, E. P. The Global Ant Biodiversity Informatics (GABI) database: synthesizing data on the geographic distribution of ant species (Hymenoptera: Formicidae). *Myrmecol. News* 24, 83–89 (2017).
- Wong, Mark KL, Evan P. Economo, and Benoit Guénard. The global spread and invasion capacities of alien ants. *Curr. Biol.* 33, 566-571 (2023).

Lines 331-2: Incomplete sentence.

Response: Thanks, we've revised as below (lines 336-339):

Although the detailed topology of the phylogeny at the species-level was still limited, the phylogeny we used here was appropriate for species-level dissimilarity analysis^{13,15,86,87,88}, because as the pairwise dissimilarity (for details see *Turnover* section) is not sensitive to the topological uncertainty in phylogeny^{89,90,91}.

Fig. 4 Legend, last sentence: Again, I think Supp Fig. 12 is not the correct figure to reference here.

Response: Thank you for pointing out this mistake, it should be Fig. 13. We've revised.

REVIEWERS' COMMENTS

Reviewer #2 (Remarks to the Author):

All revisions look good to me and I have no further comments.